# An Overview of the Application of Systems Biology in an Understanding of Chronic Rhinosinusitis (CRS) Development

**DOI:** 10.3390/jpm10040245

**Published:** 2020-11-26

**Authors:** Michał Michalik, Alfred Samet, Agnieszka Dmowska-Koroblewska, Adrianna Podbielska-Kubera, Małgorzata Waszczuk-Jankowska, Wiktoria Struck-Lewicka, Michał J. Markuszewski

**Affiliations:** 1Medical Center MML, Bagno 2, 00-112 Warsaw, Poland; michalim@mml.com.pl (M.M.); dr.alfredsamet@gmail.com (A.S.); agnieszka.dmowska@mml.com.pl (A.D.-K.); adrianna.podbielska@mml.com.pl (A.P.-K.); 2Department of Biopharmaceutics and Pharmacodynamics, Medical University of Gdańsk, Hallera 107, 80-416 Gdańsk, Poland; malgorzata.waszczuk-jankowska@gumed.edu.pl (M.W.-J.); wiktoria.struck-lewicka@gumed.edu.pl (W.S.-L.)

**Keywords:** genomics, transcriptomics, proteomics, metabolomics, chronic rhinosinusitis

## Abstract

Chronic rhinosinusitis (CRS) is an inflammatory disease of the paranasal sinuses. It is defined as the presence of a minimum of two out of four main symptoms such as hyposmia, facial pain, nasal blockage, and discharge, which last for 8–12 weeks. CRS significantly impairs a patient’s quality of life. It needs special treatment mainly focusing on preventing local infection/inflammation with corticosteroid sprays or improving sinus drainage using nasal saline irrigation. When other treatments fail, endoscopic sinus surgery is considered an effective option. According to the state-of-the-art knowledge of CRS, there is more evidence suggesting that it is more of an inflammatory disease than an infectious one. This condition is also treated as a multifactorial inflammatory disorder as it may be triggered by various factors, such as bacterial or fungal infections, airborne irritants, defects in innate immunity, or the presence of concomitant diseases. Due to the incomplete understanding of the pathological processes of CRS, there is a continuous search for new indicators that are directly related to the pathogenesis of this disease—e.g., in the field of systems biology. The studies adopting systems biology search for possible factors responsible for the disease at genetic, transcriptomic, proteomic, and metabolomic levels. The analyses of the changes in the genome, transcriptome, proteome, and metabolome may reveal the dysfunctional pathways of inflammatory regulation and provide a clear insight into the pathogenesis of this disease. Therefore, in the present paper, we have summarized the state-of-the-art knowledge of the application of systems biology in the pathology and development of CRS.

## 1. Introduction

Chronic rhinosinusitis (CRS) is an inflammatory disease that involves the accumulation of mucosa in the nose and sinuses. It can often lead to allergic rhinitis (AR), which significantly reduces a patient’s quality of life [1,2,3,4]. The composition of nasal mucus is recognized as the first barrier of the human immune system. This barrier, if functional, protects against allergens, bacterial infections, or mucociliary impairment. CRS is a multifactorial disease, with different underlying etiologies and pathomechanisms such as bacterial or fungal infections, airborne irritants, or defects in innate immunity. In addition, CRS can be triggered by the presence of concomitant diseases, such as asthma, cystic fibrosis (CF) or gastroesophageal reflux disease, chronic obstructive pulmonary disease, or aspirin-exacerbated respiratory disease [5,6,7,8]. Since the pathogenesis of CRS is not fully understood, many studies focus on different areas associated with this disease, ranging from the identification of infected sinus fluid composition to genetic screening of patients. As observed in the case of acute rhinosinusitis, the inflammatory cells predominantly seen in the sinus fluid of CRS patients are neutrophils. However, a small number of eosinophils, mast cells, and basophils may also be found. The presence of cytokine proteins, including IL-3, IL-4, IL-5, IL-8, and GM-CSF (granulocyte-macrophage colony-stimulating factors), has been demonstrated in tissue homogenates [8,9,10,11,12,13]. Moreover, due to the rapid development of new technologies and advanced computational methods, complex, multifaceted research problems can possibly be solved.

Systems biology uses a holistic approach to study the complex interactions in biological systems based on the analysis of genome, transcriptome, proteome, and metabolome [5]. In addition to the development of genomics, intensive progress has been made in the analysis of transcriptome and proteome, because RNA and proteins, as gene expression products, are directly related to them. In addition, the observed metabolites’ level reflects the consequence of dynamic changes in the genome, transcriptome, and proteome as well as the current phenotype of a particular organism. Therefore, the metabolome is considered to be related to the genotype-to-phenotype gap [14]. In the present review, we focus on each branch of systems biology in terms of its application in CRS development. Currently, numerous genetic studies have been conducted on CRS, including those related to single-nucleotide polymorphisms (SNPs) [15,16], genome-wide association studies (GWASs) [16], and pooling DNA studies [17,18]. Regarding the analysis of proteome, various potential biomarkers have been proposed for a better understanding of the normal physiology and pathogenic mechanisms of human diseases. Most of the useful information on the genes resides in the proteome, which is the sum of multiple dynamic processes including protein phosphorylation, protein trafficking, localization, and protein–protein interactions. Proteomics unravels the biological complexity encoded by the genome at the protein level by using technologies to analyze large numbers of proteins in a single experiment. Generally, proteomic research has two main facets, namely expression proteomics and targeted proteomics. Expression proteomics aims to catalog the proteome—i.e., the full complement of proteins expressed by the genome in any given cell, tissue, or body fluid at a given time. Simultaneously, targeted proteomics determines the cellular functions of the genes directly at the protein level (protein–protein interactions, post-translational modifications, protein localizations within cells, etc.). A wide range of proteomic approaches is available. These include the gel-based applications such as one-dimensional and two-dimensional polyacrylamide gel electrophoresis (2D-PAGE) and the equally accessible gel-free high-throughput screening technologies, such as multidimensional protein identification technology, isotope-coded affinity tag, stable isotope labeling with amino acids in cell culture, and isobaric tagging for relative and absolute quantitation (iTRAQ). Shotgun proteomics and fluorescence two-dimensional difference gel electrophoresis, as well as protein microarrays, are applied to obtain an overview of protein expression in biological samples. Large-scale Western blot assays, multiple reaction monitoring (MRM) assays, and label-free quantification of high-mass resolution liquid chromatography (LC)–mass spectrometry (MS) data are being explored for high-throughput analysis. Acquiring data with these new techniques is indeed a very complex multistep process that leads to new challenges in data processing and analysis. Many different bioinformatics tools have been developed to aid research in this field, such as optimizing the storage and accessibility of proteomic data or statistically ascertaining the significance of protein identification [19,20]. Based on metabolomic studies, the metabolic fingerprinting strategy [21], metabolic profiling, and quantitative targeted metabolomics are identified as useful in CRS development [22,23]. Each of the presented studies has utilized different strategies as well as specific, modern, highly sensitive analytical instrumentations. Additionally, the analysis of the obtained data sets requires comprehensive libraries and advanced bioinformatics methods. It is necessary to merge, compare, and assess such large and various data from different branches of systeomics, which is still a challenging task. To the best of our knowledge, analysis of such integrated data in CRS development has not been performed thus far. Therefore, in the present review, we summarize the current state-of-the-art knowledge of the application of systems biology in CRS, separately for genomic, transcriptomic, proteomic, and metabolomic studies.

## 2. Genomic and Transcriptomic Factors of the Disease

The development of CRS can be influenced by numerous factors, of which genetic predisposition is the leading one. According to the current literature, some evidence emphasizes the association between CRS and mutations in the cystic fibrosis transmembrane regulator (*CFTR*) gene, impairing respiratory mucosal function [24,25]. Some studies have also assessed the familial risk of CRS diagnosis in the first- and second-degree relatives [26].

In addition, the CRS genetic background can be linked with the genes associated with bacterial colonization, immunological processes in the body, primary ciliary dyskinesia (PCD), or taste receptor (T2R) genes (*TAS2R38*). Genetic strategies are usually applied as hypothesis-driven approaches or in GWAS. The first approach relies on a known a priori candidate gene apparently associated with the diseased state or theoretically involved in its pathogenesis. This approach compares the allele frequencies of SNPs in the studied group vs. controls [15,27]. It is performed on individual patients with the sequence of monogenic disorders or on families or homogeneous populations using genome-linkage studies. The second strategy, GWAS, is hypothesis-independent and identifies novel SNPs that differ in allelic frequency between the studied groups. It relies on mapping the regions of the human genome that may be associated with the disease and utilizes a high-throughput genotypic library that includes approximately 1 million SNPs [16]. Although such a study does not require the use of a homogeneous population, it requires large sample sizes (thousands of samples), which significantly increases the research costs. An alternative of GWAS that effectively reduces the study cost is the pooling GWAS (pGWAS) approach [17,18]. In the pGWAS strategy, instead of genotyping each subject individually and then calculating allelic frequencies, the genotyping study is performed on a pool of an equal amount of DNA extracted from each subject separately in the studied and control groups. Such pooled DNA is hybridized in high-throughput chips and provides estimated allele frequencies, which ultimately enables the selection of candidate genes. Subsequently, these candidates are individually genotyped.

Wang et al. [27] assessed if mutations in the *CFTR* gene responsible for CF can predispose a person to CRS. The authors analyzed the genomic DNA samples extracted from the blood of 147 CRS adult patients and 123 controls to screen them for 16 common CF mutations. The participants (*n* = 11) in whom common CF mutations were found were excluded from further study. In addition, the authors detected *CFTR* variants (5T, 7T, 9T) of the splice acceptor site in intron 8 and F508C, 1507V, and 1506V polymorphisms of the *CFTR* gene. Moreover, the *CFTR* gene exons and flanking introns were analyzed by denaturing gradient gel electrophoresis. The obtained data were statistically analyzed using univariate statistics (Student’s *t*-test and Chi-square test). Consequently, the authors found that 9 out of the 10 CF carriers had the M470V polymorphism. Moreover, M470V homozygotes were over-represented in the 136 CRS patients vs. controls (*p* = 0.03). Therefore, the authors concluded that mutations in the *CFTR* gene could predispose to CRS. Furthermore, the association between the *CFTR* gene and CRS was proposed by Pinto et al. [16]. Using the GWAS strategy, the authors found that the region of chromosome *7q31.1*–*7q32.1* related with the *CFTR* gene can be associated with the development of CRS. Other studies also confirmed the observed increase in the frequency of *CFTR* mutations in CRS patients, suggesting that populations with CRS are at a higher risk of being the carriers of a *CFTR* mutation [28,29]. Therefore, scientists are attempting to use this association in developing a new therapeutic option for treating CRS owing to the expanding knowledge of CFTR gene mutations and their presence in CRS. One example of such an application is the therapy that targets *CFTR*-mediated chloride transport to improve mucociliary clearance in CRS patients [30]. Another recently developed gene target therapy for CRS involves the use of ivacaftor (a medication approved by the Food and Drug Administration—FDA) which targets the G551D-*CFTR* mutation. With the use of this agent, the authors observed the medical reversal of chronic sinusitis in CF patients [31].

In terms of hypothesis-driven theory, there is also evidence that PCD, the autosomal recessively inherited disease affecting the structure and function of cilia, can lead to CRS [32]. PCD also leads to various pulmonary diseases and is genetically heterogeneous. In addition, variability in impaired functions and structure of cilia was observed among PCD patients. Therefore, it is challenging to use molecular genetic tests for comprehensive diagnosis in patients, as well as for the development of new target avenues. Marshall et al. [33] used whole-exome sequencing as a potential tool for patients with suspected PCD. The study was carried out on 20 previously genetically undiagnosed families, among which PCD was diagnosed in 11. Although some molecular genetic tests are currently performed for patients with PCD, the search for potential genes that could restore ciliary functions and provide some relief to CRS patients is still a viable research topic.

Some investigations concern the association of bitter taste receptors (T2Rs) and CRS. Bitter taste receptors are located in the human tongue and in the bronchial and sinonasal epithelium. When stimulated with bitter agonists, these receptors increase the intracellular calcium level, leading to increased production of nitric oxide. Such a cascade of reactions increases mucociliary clearance and induces an antibacterial effect, ultimately bringing significant relief to CRS patients [34]. In their study, Bufe et al. [35] focused on specific SNPs in the *TAS2R38* gene at positions 49, 262, and 296, which can encode functional amino acids (proline, alanine, and valine called PAV variant) and nonfunctional ones (alanine, valine, and isoleucine, termed AVI variant). Patients with PAV/PAV, PAV/AVI, and AVI/AVI genotypes were characterized as supertasters, intermediate, and nontasters, respectively. The supertasters group can effectively produce nitric oxide, have faster mucociliary clearance, and exhibit a more potent bactericidal effect compared to persons with PAV/AVI and AVI/AVI genotypes [29]. Therefore, the supertasters group was believed to experience fewer sinus infections and reveal a better sinonasal quality of life, which is taken into account concerning CRS pharmacotherapy and patients’ recovery after sinus surgery [36]. Additionally, the *TAS2R38* gene may be a future therapeutic target to promote T2R38 innate immune response in each subgroup of CRS patients.

Regarding immunological genes, numerous studies have been performed focusing on the association between the studied immunological gene and CRS, among which the human leukocyte antigen (*HLA*) should be underlined. *HLA* alleles such as *HLA-A_24*, *HLA-A_74*, *HLA-B_54*, *HLA-B1_3*, *HLA-B1_08*, *HLA-B_07*, *HLA-B_57*, *HLA-Cw_04*, *HLA-Cw_12*, *HLA-DRB1_03*, *HLA_DRB1_04*, and *HLA-DQB1_03* have been described in the literature as associated with CRS, mainly with its polyposis subtype [37,38,39,40]. Another theory states that the impairment of the innate immune system contributes to CRS since it is the first line of defense against various infections [41]. Moreover, many Toll-like receptors and the related signaling molecules such as NF-kB have been studied and assessed as differentially expressed in CRS tissue compared to the healthy control group [42,43]. The other group of immunological genes are those involved in the metabolism of arachidonic acid, genes for cytokines, or other proinflammatory genes [44,45,46]. In addition, another study searched for polymorphisms in the nitric oxide synthase gene (*NOS2A*). The authors found that polymorphisms in the *NOS2A* gene were statistically significantly associated with the eosinophilic form of CRS with nasal polyps (CRSwNP) [47].

As was mentioned earlier, instead of searching for SNPs in a known a priori gene, which is suspected to be related to the disease state, GWASs and their alternatives, pGWASs, have been successfully performed.

In the work of Tournas et al. [17], a pGWAS was performed to identify susceptibility genes for CRS with a special emphasis on the *p73* gene. In total, 206 CRS patients and 196 controls enrolled in the study. The DNA from CRS patients was extracted from the peripheral blood leukocytes, while in the case of controls, DNA was extracted from the blood (*n* = 30) or saliva samples (*n* = 166). The samples were pooled separately for CRS and control groups and subsequently genotyped by high-density SNP genotyping study. The pooled DNA was hybridized and interrogated with 555,175 SNPs. Subsequently, a subset of highly ranked SNPs most likely associated with CRS was selected for individual genotyping, which was carried out on 1536 SNPs, among which two SNPs from the *p73* gene were evaluated (rs3765731 and rs3765692). In addition, the authors sequenced the entire *p73* gene in 10 patients and 1 control. This sequencing did not identify any mutations capable of altering amino acid content in the gene product. Statistical analyses showed that one SNP from gene *p73* (rs3765731) was significantly associated with CRS (*p* < 0.05). The authors noted a significant difference in the minor allele frequency between the studied groups. The minor allele A was more frequently identified in the control group, while the minor allele G was more associated with CRS. Additionally, it was found that the homozygous AA patients had a seven-fold lower risk of CRS compared to GG homozygotes, and the minor allele A apparently had a protective effect in comparison with allele G. Further studies are also required to evaluate the involvement of *p73* gene in some intracellular signaling to effectively explain CRS development.

A pGWAS was also performed by Cormier et al. [18]. The authors extracted DNA samples from 408 CRS patients and 190 controls. Using pGWAS, they compared the DNA pools from patients with and without *Staphylococcus aureus* colonization by high-density genotyping that interrogated one million SNPs. The 39 SNPs that were most likely associated with bacterial colonization were selected for further individual genotyping. Among them, 23 SNPs were significantly associated with *S. aureus* colonization (*p* < 0.05). In total, 12 SNPs were associated with an increased risk for *S. aureus* colonization, and 11 SNPs were correlated with a protective effect against this bacterium. The authors also used the Ingenuity Pathway Analysis (IPA) software to evaluate if genes mapping close to the SNPs mentioned earlier belonged to the same functional pathway. The analysis showed that the top biological functions were related to cell-to-cell signaling, cell morphology, growth, and proliferation.

Moreover, it was found that the top canonical pathways were related to IL-8 and IL-3 signaling, micropinocytosis signaling, and the mammalian target of rapamycin signaling. The authors observed that selected SNPs were mapped within or close to 21 genes including *RYBP*, *RAC1*, *FAM79B*, and *CACNA2D1*. These genes are known to be involved in endocytic internalization and bacterial recognition. Therefore, the authors suggested that further studies on these genes and their biological functions are required as they would be promising in the search for novel strategies for CRS therapy.

Summing up the current knowledge on the genetic susceptibility to CRS, we can assume that either identification of SNPs among the known genes or search for new SNPs using a GWAS strategy would lead in the future to new target avenues of CRS and aid in better understanding the disease pathogenesis. Table 1 summarizes the current genetic variants found to be associated with CRS.

## 3. CRS in the Field of Proteomics

Analyses of protein components in biological fluids, such as the blood (serum and plasma), urine, saliva, cerebrospinal fluid, and nasal mucus, can increase the current knowledge on the biology and physiology of the conditions of human body [43,44,45,46,48,49,50,51]. Diseases cause alterations in the production, secretion, and excretion of proteins from the affected tissues or organs into the body fluids. Qualitative and quantitative analyses of proteome samples derived from complex biological mixtures are important to fully understand the functions of cell biology. The term proteome refers to all proteins produced by a species; similarly, genome is the entire set of genes. Unlike the genome, the proteome varies with time and has been defined as the proteins present in each sample (e.g., tissue, organism, cell culture) at a certain time point. The main components of a proteomic study include: (i) sample collection and protein extraction, (ii) protein or peptide fractionation, (iii) detection of peptides or proteins, and (iv) interpretation and protein identification. A proteomic study of body fluids, including nasal mucus, is complex and involves multiple steps. The examination of nasal lavage fluids (NLFs) aids in monitoring the alterations caused by disease states of the airways such as sinusitis, seasonal AR, CF, asthma, or general respiratory disorders. Nasal lavage is a simple sampling of the nasal fluid to investigate disease-related cellular and solute compositions [49,52,53,54]. In proteomic studies, NLFs can be a source of information about proteins involved in acquired and innate immune responses produced as a result of microbial infections and inhalation of unclean air.

In addition, the improvements in proteomic technology could lead to the search for safer, more effective, and new drugs. Casado et al. [55] compared the protein profiles of NLFs from subjects with sinusitis before and after stimulating glandular secretion as well as after pharmacological treatment. The protein identification was performed using capillary liquid chromatography–electrospray-quadrupole time-of-flight mass spectrometry (CapLC-ESI-Q-TOF/MS). This study identifies the complex protein profiles present in acute sinusitis nasal secretions as well as proteins that respond to pharmacological treatment. According to their biological functions and origin, a number of proteins were found in sinusitis NLFs pre- and postpharmacological treatment. The examination of prepharmacological treatment NLFs demonstrated the existence of 9 glandular proteins, 7 inflammatory proteins, 10 regulatory proteins, 1 neuronal protein, 29 plasma proteins, 12 cellular proteins, and 10 proteins (unknown). On the other hand, in the case of postpharmacological treatment NLFs, the number of proteins was found to be 3 glandular proteins, 3 regulatory proteins, 6 plasma proteins, 2 cellular proteins, and 4 unknown proteins. The proteomic approach hypothesized that the treatment the subjects had received was successful and blocked the influx of inflammatory cells, the generation of their mediators, and the consequent vascular permeability and glandular hypersecretion.

A comprehensive examination of NLF from 10 volunteers, collected before and after they had been submitted to nasal provocations, was performed by Casado et al. [56] using the CapLC-ESI-Q-TOF/MS technique. They identified a total of 111 proteins, 42 of which have never been previously detected in NLF. The authors suggested that the detected polypeptides were involved in innate (27%) and acquired immunity (21%) systems and were the major components of NLF. The deleted in malignant brain tumors 1 isoform precursor and cytoskeletal proteins were identified with a high statistical score. In addition, three proteins of the palate lung nasal epithelial clone (PLUNC) family—including short-PLUNC1 (SPLUNC1), long-PLUNC1 (LPLUNC1), and long-PLUNC2(LPLUNC2)—were identified. Cellular components (52% of all proteins identified) such as cytoskeletal (33%), functional (15%), and regulatory (4%) proteins, normally present in the nasal cavity, have also been identified. Benson et al. [57] provided an extensive list of NLF proteins generated from CRS patients with coexisting asthma. The authors used a combination of affinity, anion exchange, and reverse-phase chromatography prior to the LC–MS/MS analysis to detect a lower abundance of proteins in intact NLF. The samples were fractionated using anion exchange chromatography followed by reverse-phase LC and digested with trypsin before the LC–MS/MS analysis. Owing to this study, the authors compiled lists of 197 NLF proteins identified from patients with nonpolypoid AR or CRS with coexisting asthma. Although the validity of directly comparing the two protein lists obtained from two groups of samples of AR and CRS NLF is unclear, some differences were observed. Five proteins, a-2-macroglobulin, actin, long palate, lung and nasal epithelium carcinoma-associated protein, myosin 1, and myosin 4, were identified from AR NLF with 16, 19, 13, 12, and 25 unique peptides, respectively; however, these were not observed in the asthmatic CRS NLF. Similarly, lipocalin and lactotransferrin were identified from AR NLF with 29 and 47 peptides, respectively, compared to the CRS NLF identified with 3 and 11 unique peptides. In contrast, semenogelin 1 and 2 were identified in asthmatic CRS NLF with 15 and 6 unique peptides, respectively; however, these were not observed in AR NLF. An overview of the current literature suggests that NLF protein profiles differ from conventional samples of, i.e., whole blood; hence, it is capable of complementing or even expanding the biomarker index thus obtained. NLF is a readily available and noninvasive sample. However, it is associated with significant problems such as sample contamination. For example, the presence of keratins in samples is often due to contamination during sample preparation. This protein can be present at high levels in NLFs due to the airway mucosa environment. In addition, the presence of plasma proteins, or more likely, the denaturation or degradation of these proteins in one or more individual CRS NLF samples can generate falsified results. Issues such as blood serum leakage and high salt content in the samples of NLF were reported by Schoenebeck et al. [58]. The authors proposed a modified protocol for NLFs analysis as an appropriate method to reproducibly detect typical NLF components with a concomitant decrease in blood proteins. They determined protein profiles of NLFs using analytical techniques, such as 2D-PAGE, followed by protein digestion and LC–MS/MS analysis. In their work, protein profiles from the NFLs were efficiently detected at a significant level owing to the prewashing step of both nostrils, which reduces the impurities derived from mucosal bacteria. In addition, the authors recommended collecting NLF samples of saline concentrations below 10%, since enhanced salt concentrations would falsify the innate composition of NLF. A different study conducted by Kim et al. [12] analyzed the proteomes from nasal secretions collected on a filter paper. The authors identified a certain number of proteins (2020 proteins in nasal secretions) involved in the known biochemical pathways. Such a large set of proteins, compared to other studies, was identified due to the fact that the nasal secretions were retained on the filter paper. This type of sample collection could provide a solid protein-stabilizing matrix that prevents cytokine-mediated protein neutralization and protease degradation. Additionally, the sample preparation steps were minimized to reduce peptide loss. The analyses were performed using LC–MS detection with a quadrupole Orbitrap mass spectrometer in data-dependent acquisition (DDA) and data-independent acquisition (DIA) modes. Consequently, the authors identified 2020 proteins in the DDA mode and 1278 in the DIA mode in nasal secretions. Canonical pathway analysis and a gene ontology (GO) evaluation revealed that IL-7, IL-9, IL-17A, and IL-22 signaling and neutrophil-mediated immune responses such as neutrophil degranulation and activation were significantly increased in the CRSwNP group compared to the control group. The GO terms related to the iron ion metabolism may be associated with the development of CRS and nasal polyps (NPs). In sum, the collection of nasal secretions on the filter paper is a practical and noninvasive method for an in-depth study of nasal proteomics. Nasal biopsy is a highly invasive procedure and requires expertise not only in tissue sampling but also in biopsy processing. However, tissue specimens can be used to evaluate both protein and gene expression. Min-man et al. [59] used an invasive method to obtain tissue samples of the NP, chronic sinusitis, and normal nasal mucosa in order to determine new proteins in those samples compared to those detected in NLFs. Each of the samples was collected during functional endoscopic sinus surgery. The tissue samples were separated by immobilized pH 4–7 gradient 2-dimensional gel electrophoresis (2-DE). The authors identified the protein profiles (peptide mass fingerprinting) by matrix-assisted laser desorption/ionizationtime-of-flight MS. The peptide sequence data were obtained by electrospray ionization quadrupole time-of-flight MS. Consequently, 30 differentially expressed proteins among the three kinds of tissues were identified. A1–7 spots (alpha-1-antitrypsin (A1), fibrinogen gamma-A chain precursor (A3), APOA1 protein (A5), plasma retinol-binding protein (A6), transthyretin (A7)) were expressed significantly in the NP sample. B1–18 spots (PLUNC (B7), PACAP (B9), glutathione S transferase-pi (B10), NKEF-B (B12), Cu/ZnSOD (B13), DJ-1 protein (B18)) were expressed significantly in the chronic sinusitis sample. C1–5 spots were expressed significantly in the normal nasal mucosal tissue.

From inhaled air, the airways are confronted with a variety of pathogens required to be filtered and transported away. Nasal mucus is the first-line defense barrier and functions as a physical barrier against particles, irritants, microbes, viruses, food, and liquids. Hairdressers have an increased risk of developing respiratory symptoms from the airways such as asthma and rhinitis, as they are exposed to many reactive chemicals that can induce immunological reactions. Mörtstedt et al. [60] analyzed the changes of protein profiles in NLF from persulfate-challenged subjects to identify those proteins potentially involved in the pathogenesis of bleaching powder-associated rhinitis or candidate effect biomarkers for persulfate. The authors collected NLF samples from hairdressers with and without bleaching powder-associated rhinitis and analyzed them using LC–MS/MS and selected reaction monitoring (SRM) mode targeting 246 NLF proteins and five oxidized peptides. Samples collected from hairdressers with bleaching powder-associated rhinitis were compared to those obtained from asymptomatic hairdressers and atopic subjects without work-related exposure to persulfate. Consequently, a list of nine proteins seemed to be affected by the persulfate challenge and should be followed up was defined and is shown in Table 2. In addition, the levels of five oxidized peptides were measured in the samples using LC–SRM-MS/MS technique, and an albumin peptide containing oxidized tryptophan was found to be increased at 2 and 5 h after the challenge. This oxidation may be the result of endogenous oxidative stress, which suggests that such peptides might serve as biomarkers of oxidative reactions in the future.

For a better understanding of the molecular mechanisms underlying respiratory diseases, Biswas et al. [61] correlated sinus mucus proteins with disease state, host inflammatory cell response, and associated bacterial diversity. Among other tools, they used proteomics, specifically iTRAQ, to identify proteins in the mucus collected from the middle meatus of the CRS patients and healthy individuals. Of the total 606 proteins identified in this study, seven were highly (*p* < 0.05) abundant and 104 were significantly lower in the CRS cohort compared to healthy controls. Table 2 shows the list of significantly elevated proteins in the samples of CRS patients. Another study conducted by Upton et al. [62] investigated the differences in protein abundances within the sinonasal mucosa of the CRSwNP patients compared to healthy controls using 2-DE. The highly abundant proteins were detected by MS. Eight proteins were found to be significantly increased in the mucosa of patients with NP compared to controls, whereas seven proteins were significantly decreased (Table 2). Nasal mucus has been studied using a survey or shotgun proteomic approach, in which as many compounds as possible were detected. Tewfik et al. [63] evaluated the qualitative and quantitative differences in the protein content of the nasal mucus in patients with chronic hypertrophic sinusitis with nasal polyposis compared to control subjects. The authors noted that the content of cytokines, eosinophils, immunoglobulins, and immune mediators differed in the nasal and sinus mucosa of CRS patients. Mucus hypersecretion in CRS occurs as a result of hyperplasia and hypertrophy of nasal acinar cells. The proteins in samples were digested using proteolytic enzymes, labeled with iTRAQ reagents, separated based on 2-dimensional capillary LC and subjected to MS detection. In total, 35 proteins were identified in their study, and some of them are presented in Table 2. The analysis of such a complex sample is challenging due to the full range of protein concentrations present. An important consideration in such research is the influence of a biological material collection step on the obtained protein profile. Suctioning allows a directed and straightforward collection of representative nasal mucus. The relative lack of vascular proteins in the sample indicates a lack of vascular trauma and consequent contamination. The iTRAQ-based studies are notoriously expensive and time-consuming if they deal with a large number of proteins and samples. A quantitative proteomic approach using the MRM mode has been developed with both high sensitivity and selectivity, but at the cost of restricting the discovery of new potential proteins. Based on a previous pilot study conducted by Tewfik et al. [63], Badaai et al. [64] continued this research by using MS in MRM mode to investigate the quantitative differential expression of specific nasal mucus proteins in CRSwNP patients compared with healthy subjects. Initially, qualitative observations from the analysis by sodium dodecyl sulfate-polyacrylamide gel electrophoresis indicated a marked protein banding difference in the CRS group compared to controls.

They identified 7 of the 10 targeted proteins in the MRM mode, many of which were related to innate and acquired immunity. Quantitative analysis showed differential expression of some proteins in CRS patients compared to the control subjects, which included lysozyme C, IGLV4-3 (immunoglobulin lambda variable 4-3), bactericidal protein, calgranulin, lipocalin, mucin B, and deleted brain tumor protein (Table 2). The pathogenesis of CRS reflects a chronic inflammatory process of the paranasal and sinus mucosa, which is characterized by the overproduction of mucus. Since mucin glycoproteins are the primary macromolecular components of mucus and other mucosal parts, they act as a physiological barrier against contaminants and bacteria by forming a viscoelastic mucus gel. Saieg et al. [65] verified the hypothesis that whether sinonasal secretions from CRS pediatric patients would contain MUC5B in overabundance relative to other mucin glycoproteins. Using label-free proteomic techniques, they created a protein profile showing the varying expression of 294 proteins in sinonasal secretions from CRS pediatric patients and controls. Many of the identified proteins with significant fold changes in the CRS sinonasal secretions were involved in the innate immune response. Saieg’s group noticed that several intrinsic immunity proteins such as DMBT1, S100A8, S100A9, MYH9, BPIFA1, LTF, and IGHA1 exhibited an increased expression in the samples of CRS secretions and varied in their levels in CRS NLF or sinus tissues.

Interestingly, DMBT1 protein is closely associated with MUC5B in the respiratory tract secretions. Based on the reviewed literature, many proteins that are significantly elevated in the CRS sinonasal secretions are implicated in the airway inflammatory reaction and immune response [65]. Some of them might be useful for further investigation as potential biomarkers for CRS. In addition, as shown by various studies, due to the high sensitivity and versatility of available proteomic methods, the easy-to-collect nasal discharge is an advantageous material to continue the search for biomarkers.

## 4. Metabolomics in Searching for CRS-Related Metabolites

Metabolomics aims at qualitative and quantitative analysis of low-molecular-weight endogenous compounds (up to 1500 Da) in biological samples (urine, blood, tissue extracts, or sinus mucosa). The observed levels of metabolites result from the dynamic changes in the genome, transcriptome, and proteome. In addition, they reflect the current phenotype of a particular organism. Therefore, it is considered that the metabolome is related to the genotype-to-phenotype gap [66,67]. Three main strategies are commonly applied in metabolomics, namely targeted metabolomics, metabolic fingerprinting, and metabolic profiling [68,69,70]. Metabolic fingerprinting denotes a qualitative analysis focused on identifying as many metabolites as possible in biological samples to detect some significant changes in their relative levels among the studied groups (e.g., diseased patients vs. healthy controls, patients with the preliminary stage of the disease vs. those with advanced stage, patients with severe cancer vs. those with remission, etc.). This would aid in explaining the pathogenesis of the disease or indicate its presence or progression. To detect a broad set of metabolites characterized by different physicochemical properties, various complementary analytical techniques are usually applied, such as gas chromatography (GC), capillary electrophoresis (CE), and high-performance LC, all coupled with MS detection as well as a nuclear magnetic resonance (NMR) technique [71]. The sample pretreatment procedure is then usually limited only to a few steps in order to not lose the potentially essential compounds. These steps include deproteinization (plasma, sinus mucosa), homogenization (tissue), dilution (urine), centrifugation, and filtration (all biological samples). In addition, sample pretreatment involves an additional procedure, the type of which depends on the analytical technique used. For example, in the GC technique, the extra step relies on the derivatization of nonvolatile compounds into volatile forms. The NMR technique requires the dilution of samples in deuterated water. In a metabolic fingerprinting study, which is also called an untargeted approach, vast amounts of data sets are required to be analyzed using advanced bioinformatics tools to select potentially relevant compounds in the studied disease. This includes a laborious data pretreatment procedure (filtration, peak alignment, normalization, and scaling) as well as uni- and multivariate chemometrics or hierarchical Bayesian modeling [72,73]. Consequently, the metabolites that are most varied in their levels among the studied groups are selected. Such an untargeted approach of finding “a needle in a haystack” should be supported by a targeted metabolomic study. This strategy relies on the quantitative analyses of selected metabolites after complete method validation [74,75]. The selected metabolites can be chosen based on the results of the untargeted study and the available literature or state-of-the-art knowledge of the disease pathogenesis. Since, in a targeted study, the known set of metabolites is determined, the sample pretreatment procedure is usually composed of more steps compared to untargeted ones. It can include extraction methods such as liquid–liquid extraction (LLE), solid-phase extraction (SPE), and solid-phase microextraction, which allow samples to be properly cleaned and more concentrated. Moreover, the use of complementary analytical techniques is not required. Depending on the chosen metabolites, either a LC or GC or CE technique alone is used. The samples are analyzed by the developed and validated method. Such a validation approach could be utilized in different laboratories to guarantee its quality and repeatability [76].

The data set obtained from a targeted study is a matrix composed of only selected metabolites vs. analyzed samples, which is a much simpler one compared to the matrix acquired in an untargeted approach. Therefore, the pretreatment and analysis of data are less laborious and a similar type of statistical method is used for both approaches. As a result of the targeted metabolomic study, practically only a few out of previously chosen metabolites appear relevant and statistically significant among the studied groups [76]. Therefore, such confirmation is required in the procedure of searching for potential disease-related compounds. The last metabolomic strategy is called metabolic profiling. It focuses on analyzing the metabolites that belong to the same chemical group such as amino acids, nucleosides, carbohydrates, purines, carnitines, and long or short fatty acids [77,78]. This strategy is usually an untargeted one; however, there are also many reasons for selecting the quantitative determination of the selected metabolites derived from the same chemical origin. Metabolite profiling can be used to identify the metabolites that build common biochemical pathways (e.g., RNA degradation, purine biosynthesis, long-chain fatty acids oxidation, Krebs Cycle), which could be significantly weakened or enhanced during disease progression. Gaining more profound knowledge about the altered signaling of biochemical pathways can be advantageous in explaining disease pathogenesis. Another strategy defined as metabolic footprinting relies on an untargeted analysis of metabolites excreted from, e.g., bacteria to the culture media [79]. However, to the best of our knowledge, no data are found in the literature on the bacteria metabolomics (both fingerprinting and footprinting) isolated from patients with CRS.

Fazlollahi et al. [21] demonstrated the application of an untargeted lipidomic study for the analysis of lipids among other metabolites. The authors assessed sinonasal tissue samples (sinus mucosa) from 9 patients with CRS with concomitant NP, 11 patients with CRS without polyps, and 12 healthy controls. Since the study aimed to analyze and identify various classes of lipids, the lipids selective modified Bligh-Dyer LLE was used as the sample pretreatment method. The primary analyses involved gas chromatography/electron impact-mass spectrometry (GC/EI-MS) and flow-injection/electrospray ionization-tandem mass spectrometry (FI/ESI-MS/MS). For the GC-MS study, the samples were subjected to additional derivatization steps. The authors used pooled sinonasal tissue samples separately from each group to determine major lipid classes and evaluate the general differences in their content among the studied groups. From GC/EI-MS analysis, the authors identified cholesterol and fatty acids such as palmitic, oleic, stearic, and lauric acids in the analyzed samples. The cholesterol signal was overloaded, and therefore it was impossible to compare its level among the studied groups. The levels of fatty acids were, on average, 10-fold higher in CRS patients with concomitant NP growth compared to other groups, which suggested an altered lipid metabolism in the sinus mucosa samples. The FI/ESI-MS/MS technique confirmed the presence of different lipid species such as phosphatidylcholines, phosphatidylethanolamines, ceramides, and cholesteryl esters. Cholesteryl palmitoleate was present in CRS patients with concomitant NP, whereas in the case of those with CRS without nasal polyps (CRSsNPs), it was not detectable; cholesteryl arachidonate was present in all CRS sinus mucosa samples, but not in NP. This compound, along with cholesteryl linoleate, would be a potential antimicrobial lipid agent with microbicidal effect against *Pseudomonas aeruginosa*, the pathogen responsible for various inflammation processes [80]. Further quantitative analysis of these cholesteryl esters across different sinonasal tissue samples can be performed in CRS patients.

In the study conducted by Miyata et al. [22], a multiomic (transcriptomics, proteomics, and metabolomics) approach was proposed for the isolation of CD69hi CCR3low CXCR4-siglec-8int eosinophils from NP from patients (*n* = 6) with eosinophilic rhinosinusitis (NP-EOS-nasal polyp-derived eosinophils) and healthy controls (*n* = 6, PB-EOS- peripheral blood-derived eosinophils). Using the metabolomic approach, the authors performed a targeted lipidomic study with LC-MS/MS. Briefly, the isolated eosinophils were suspended in Hank’s buffer salt solution, incubated, and centrifuged. The cell supernatant was diluted in methanol (1:1, *v*/*v*) and extracted using anion-exchange SPE columns after the addition of deuterated internal standards. The targeted analyses of various eicosanoids and docosanoids were carried out using an Acquity UPLC™ BEH C18 column (1.0 × 150 mm, 1.7 μm) on the triple quadrupole linear ion trap mass spectrometer in negative ionization and MRM modes. In the analysis, the authors observed significantly lower amounts of both cyclooxygenase (COX) and lipoxygenase (LOX) products in the NP-EOS group compared to the PB-EOS group. Regarding COX products, lower amounts of prostaglandin E2 (PGE_2_), prostaglandin D2 (PGD_2_), thromboxane B2 (TXB_2_), 12-hydroxyhepentadecatrienoic acid (12-HHT), and 11-hydroxyeicosatetraenoic acid (HETE) were observed. In the case of LOX products, lower levels were observed for 12-hydroxyeicosatetraenoic acid (12-HETE), 15-hydroxyeicosatetraenoic acid (15-HETE), lipoxin A4, 8,15-diHETE, and 17-hydroxydocosahexaenoic acid (HDoHE). The authors also noticed a selective upregulation of LTD_4_ biosynthesis in NP-EOS patients. Furthermore, despite the observed downregulation of COX and LOX pathway products in NP-EOS patients, no significant changes were noted in the level of arachidonic acid, a precursor of COX and LOX, which suggests that patients with NP-EOS may display unique lipid mediator profiles that can also affect specific EOS phenotypes.

The targeted analysis was also proposed by Tsybikov et al. [23] to assess the potential role of selected compounds as biomarkers in various types of chronic rhinitis. Along with one metabolite, three proteins were found in the serum and nasal secretions of CRSwNP (*n* = 54) and CRSsNP (*n* = 46) patients, healthy controls (*n* = 40), patients with AR (*n* = 51), and patients with nonallergic rhinopathy (*n* = 43). In a targeted metabolomic study, the authors used the immunoassay method for the determination of neopterin. Human monocytes and macrophages mainly produce this metabolite, and its amount corresponds with the capacity of reactive oxygen species production by these cells [81]. Therefore, neopterin can be treated as an indicator of Th1 polarized immune activation [82]. The obtained data sets were statistically analyzed in the mentioned study using a one-way analysis of variance followed by a post hoc *t*-test.

The levels of neopterin were significantly elevated (*p* < 0.001) in CRSsNP patients compared to healthy controls in both sera and nasal secretion samples. Furthermore, neopterin levels measured in nasal secretions were also significantly elevated (*p* < 0.001) in CRSsNP patients compared to those with CRSwNP. Such a correlation was not observed in the case of sera samples. In addition, no significant correlation was found between the neopterin levels among the other CRS groups. Therefore, this metabolite would be proposed as a potential biomarker only for CRSsNP patients. To confirm those findings, the analyses should be subsequently carried out on a larger group of patients using more sensitive analytical instrumentation involving, e.g., MS detection.

A different approach was proposed by Broza et al. [83]. The authors applied the study based on untargeted metabolomic analysis of volatile organic compounds (VOCs) emitted from body sources. Analyses of breath samples were carried out on 17 CRS patients, 24 patients with CRSwNP, and 30 healthy controls, for which the authors combined nanotechnology with a pattern recognition method. They used a cross-reactive nanoarray based on molecularly modified gold nanoparticles and molecularly modified single-walled carbon nanotubes. The study aimed to differentiate CRS and nasal polyposis because these two types of sinusitis subgroups can have different pathological processes, and therefore their diagnosis and treatment should also be differentiated. The exhaled mixed breath samples (alveolar and dead space, 750 mL) were collected and trapped in Tenax^®^-TA/Carboxen^®^-1018 sorption tubes with end fittings and subsequently stored at 4 °C before analysis. The air trapped in sorption tubes was thermally desorbed and delivered into a chamber containing nanomaterial-based sensors during the study. The authors used 11 sensors and measured a recorded reversible time-dependent change in electrical resistance during the interaction between VOCs and nanoparticle layers. The collected data were analyzed using discriminant function analysis and divided into a training set (70%) and a test set (30%) for method validation. The authors compared four binary relations and assessed both false-positive and false-negative and true-positive and true-negative results. The correlated data were as follows: CRS vs. NP, CRS vs. controls, NP vs. controls, CRS, and NP vs. controls. The accuracy of the training set was in the range of 82–84%, whereas for the validation (test) set, it ranged from 67% to 90% depending on the data compared. A sensitivity of 100% was achieved for all sick patients (NP and CRS vs. control), whereas a 100% specificity was achieved for the CRS vs. NP group. Thus, the preliminary results seemed to be promising; however, as the authors noticed, the proposed approach requires that more patients should be enrolled in the study. Furthermore, the authors did not identify the VOCs responsible for the obtained recorded change in electrical resistance, so untargeted metabolomics followed by identification should be carried out for VOCs.

Exhaled breath samples were also examined by Liu et al. [84]. The authors measured the exhaled nasal nitric oxide (nNO) to assess the presence and degree of inflammation in the airway mucosa. They analyzed breath samples from patients with CRSwNP (*n* = 54) and CRSsNP (*n* = 34) as well as from healthy controls (*n* = 20). Patients with CRS were also classified based on atopic presence. The classification of patients in the CRS subgroup was performed after nasal endoscopy examination and sinus computed tomography. In contrast, atopic presence was evaluated with a skin prick test using 21 aeroallergens. The breath samples were collected using a handheld portable NIOX device (Aerocrine AB, Solna, Sweden). The obtained data sets were evaluated using univariate statistics (independent *t*-test, multiple *t*-tests, or *U* Mann–Whitney test). The results showed that the mean nNO level in healthy controls was significantly higher than that in both groups of patients (CRSwNP and CRSsNP) (*p* < 0.001). In addition, patients with CRSsNP had a significantly higher level of nNO compared to patients with CRSwNP (*p* < 0.001). Regarding atopic subgroups, the nNO levels were markedly higher in both CRS groups of patients with atopy and nonatopic patients (*p* < 0.001). The area under the receiver operating characteristic curve for all studied groups (atopic vs. nonatopic, CRSwNP vs. CRSsNP, controls vs. each CRS group of patients) was in the range from 0.83 to 0.988 with sensitivity and specificity ranging from 0.83 to 0.95 and from 0.70 to 0.97, respectively. Therefore, the exhaled nNO level could be used to differentiate CRSwNP patients from those with CRSsNP, although the atopic status of a patient also has some influential role in changing the nNO level. To confirm such preliminary results and to evaluate the significance of the presence of atopy as well as the potential role of nNO as a diagnostic and monitoring marker, the study should be validated in a larger group of both CRS patients and healthy controls. A summary of the results from metabolomic studies is presented in Table 3.

## 5. Conclusions

Chronic sinusitis significantly reduces a patient’s quality of life. Treatment for this condition is mainly focused on combating local infection/inflammation or improving sinus drainage. There is still a need to search for new indicators directly responsible for the pathogenesis of the disease. The application of systems biology seems to be promising mainly because it concentrates on genetic, proteomic, and metabolomic levels. Due to the use of advanced analytical instrumentations, libraries, and bioinformatics methods and the examination of the vast amounts of data obtained from the analysis of the genome, transcriptome, proteome, and metabolome, the omics can be interpreted and assessed. The current genomic studies confirm the relationship observed between CRS and mutations in the *CFTR* gene, which affects the mucociliary transport impairment. Additionally, genes encoding HLAs and involved in arachidonic acid metabolism may contribute to the pathogenesis of CRS. The genetic background of this disease can also be related to the genes responsible for bacterial colonization, immunological processes in the body, PCD, or taste receptor genes. Therefore, the future direction of genomic analyses should confirm these findings on extensive, well-designed studies with detailed patient phenotyping. Moreover, other potential concomitant diseases (atopy, asthma, allergy) should be genetically studied as pathogenic mechanisms other than CRS. Furthermore, a new therapy should be developed targeting the genes of interest, such as ivacaftor, which as mentioned in this review target the G551D-*CFTR* mutation and improve mucociliary clearance in CRS patients. Besides the mentioned proteomic studies allowing to create protein profiles that can differentiate patients with CRS vs. controls, some of these proteins may be useful as potential CRS biomarkers, e.g., mucin glycoprotein, for further research. Nasal and sinus secretions in children with CRS may contain an excess of MUC5B compared to other mucin glycoproteins. The nasal mucus, as shown by various proteomic studies, could be a source of biomarkers. Among many different proteins identified as important in CRS based on the current literature (Table 2), some should be further investigated before being established as a biomarker candidate. In the case of metabolomics, lipidomic studies were carried out showing the significance of metabolites from arachidonic acid metabolism in CRS pathogenesis. These findings are in agreement with genetic studies that also emphasize the significant role of genes involved in the metabolism of arachidonic acid. Therefore, the integration of genomic and metabolomic studies that focus on arachidonic acid metabolism may bring additional knowledge of CRS pathogenesis or propose potential CRS biomarkers. Another metabolite presented in this review is neopterin, which may be a potential biomarker for patients with CRSsNP. Human monocytes and macrophages produce neopterin, and its amount determines the ability of these cells to produce reactive oxygen species. The metabolomic studies conducted confirmed the significantly elevated levels of neopterin in CRSsNP patients compared to healthy subjects. The future direction of such a targeted metabolomic study could be its validation on a large population with different stages of CRS to assess the sensitivity and specificity of neopterin as a potential CRS biomarker. Furthermore, the metabolomic studies should include analyses for examining the relationship between the concentrations of metabolites, epidemiological data (age, sex, body mass index), and basic blood biochemical parameters (morphology, glucose concentration, lipid profile), which can possibly reveal the potential biomarkers of the disease. Unfortunately, to the best of our best knowledge, no single study integrates all or any two of the omics-related fields. However, future research in CRS should concentrate on integrating knowledge from different levels of the cell organization, resulting in a better understanding of the disease pathogenesis and indicating the new directions of pharmacotherapy.

## Figures and Tables

**Table 1 jpm-10-00245-t001:** Genetic variants found to be associated with Chronic rhinosinusitis (CRS).

Gene Annotation	Chromosome Location	Phenotype	Variation Surveyed	Allele Frequency	Number of Patients	References
*CFTR*	7q31	CRS	Multiple SNPs	0.07; 0.02	147 patients with CRS and 123 controls	[27]
	7q31.1–7q32.1	CRSsNP	Multiple SNPs	N/A	8 out of 291 screened	[16]
	7q31	CRS	Multiple SNPs	0.36; 0.14	26 CRS patients, 27 controls	[28]
*TP73*	1p36	CRS	SNP: rs3765731	N/A	206 CRS patients and 196 controls	[17]
*UBE3A*	15	CRSwNP	SNP: rs1557871 SNP: rs1557874	0.11; 0.19 0.12; 0.19	408 CRS patients and 190 controls	[18]
*HLA-A*	6p21	CRSwNP with AIA	Multiple SNPs	0.278; 0.125	66 patients with CRSwNP and 100 healthy controls	[37]
*HLA-B*		CRSwNP	Multiple SNPs	Varied		
*HLA-C*		CRSwNP	Multiple SNPs HLA-Cw-12	Varied 0.167; 0		
*HLA-DR*		CRSwNP with AIA	HLA-DRB1-04	0.444; 0.234		
		CRSwNP in ATA	Multiple SNPs	Varied	467 asthmatic patients (158 NP positive, 309 NP negative)	[39]
		CRSwNP	HLA-DR-7	N/A	50 polypectomized patients, 50 healthy controls,	[40]
*HLA-DQ*		CRSwNP	Multiple SNPs	Varied	66 patients with CRSwNP, and 100 healthy controls	[37]
*IL1A*	2q14	CRSwNP	rs17561	0.44; 0.18	82 CRS patients with NP and 106 healthy volunteers without sinonasal disease	[44]
*IL1B*		CRSwNP	rs16944	0.38; 0.46		
*TNF*	6p21	CRSwNP	rs1800629	0.19; 0.12		
		CRS		0.08; 0.11	38 patients with intractable chronic sinusitis, 35 controls,	[45]
*LTC4S*	5q35	CRSwNP, CRSsNP	Multiple SNPs	Varied	206 CRS patients, and 200 controls	[46]

Abbreviations: AIA = aspirin-intolerant asthma; ATA = aspirin-tolerant asthma, CRS = chronic rhinosinusitis; CRSsNP = chronic rhinosinusitis without nasal polyps; CRSwNP = chronic rhinosinusitis with nasal polyps; N/A = not applicable; NP = nasal polyps; SNP = single-nucleotide polymorphism.

**Table 2 jpm-10-00245-t002:** Some significantly changed proteins suggested for follow-up studies.

Protein Name	Possible Function/Biological Meaning	References
Caspase-14	Associated with epithelial barrier integrity. This protein is thought to be involved in the degradation of profilaggrin into filaggrin, which is essential for the hydration of the epidermis and skin barrier functions. Mutations in the filaggrin gene are major predisposing factors for atopic dermatitis, but they are also associated with atopic asthma and allergic rhinitis.	[60]
DMBT1 Deleted in malignant brain tumors 1 protein	Mucosal defense protein. Increased expression associated with respiratory inflammation. This protein might be a local regulator of homeostasis, through linking mucosal inflammation to the modulation of epithelial regeneration. This protein is a mucin-like molecule participating in mucosal immune defense; it was overexpressed in CRS patients.	[56,63,64,65]
Desmoplakin	This protein is associated with epithelial barrier integrity. The increased levels of this protein could reflect an enhanced tissue repair or increased degradation of desmosomes after the persulfate challenge. Desmoplakin is a component of desmosomes. The exact role of this protein in persulfate-associated rhinitis needs to be further studied.	[60]
Glutathione S-transferase P	Oxidative stress defense	[60]
IL1RN	This protein inhibits the activity of IL-1α and IL-1βas well as acute and chronic inflammation. IL-1 receptor antagonist protein (IL1RN) has a protective role toward persulfate-induced rhinitis.	[60]
Mucin-5B	Mucus secretion. A balanced mucus production is important for normal lung functions, while the overproduction of mucus has been associated with many airway diseases, such as chronic obstructive pulmonary disease, chronic bronchitis, CF, asthma, and CRS. Its high expression levels may indicate the need for a high-viscosity secretion to effectively entrap microorganisms at the primary site of their entry to the nasal airway.	[63,64,65]
Peroxiredoxin-1	Oxidative stress defense	[60]
Uteroglobin	Decreased expression of this protein is associated with respiratory epithelial damage. Its lower levels in NLF are related to exposure to tobacco smoke, CRS, and allergen challenge in patients with intermittent allergic rhinitis. Uteroglobin has been suggested as a biomarker of respiratory epithelial damage in patients with acute and chronic exposure to airway irritants.	[60]
WFDC2	Decreased expression of this protein is associated with the differentiation of bronchial epithelial cells.	[60]
S100-A9	This protein is an important proinflammatory mediator that induces chemotaxis of neutrophils and monocytes. Its high abundance supports its importance in the immune response in CRS patients. Additionally, it has antimicrobial properties and can bind to essential metals, such as calcium and zinc.	[9,55,61,63,64,65]
PRRC2C (isoform 3)	This protein is found in increased abundance in the samples of CRS patients. It helps the stem cells to differentiate into myeloid or lymphoid lineages of blood cells and also combat the chronic nature of this disease.	[61]
40S ribosomal protein S12 (RPS12)	This protein is found in increased abundance in the samples of CRS patients. The role of this protein is to aid in the catalysis of protein synthesis, but its exact role in CRS requires further clarification.	[61]
Ras-related protein Rab-14 Angiotensinogen Low-affinity immunoglobulin gamma Fc region receptor Ill-B Cell division control protein 42 homolog	This protein is highly abundant in the samples of CRS patients. Presumably, if expressed before the onset of CRS symptoms, this protein could serve as a biomarker for CRS and used for diagnostic purposes or targeted drug therapy.	[51,61]
Eosinophil lysophospholipase or Charcot–Leyden crystal protein (CLCP)	Vascular protein. This protein is a unique and prominent constituent of human eosinophils that form hexagonal bipyramidal crystals seen in the tissues and secretions from the sites of eosinophil-associated inflammation. Activated eosinophils and their mediators have been shown to damage the epithelium of the paranasal sinuses through cytolytic effects.	[62]
Apolipoprotein A-1 (ApoA1)	Vascular protein. ApoA1 is the major protein component of high-density lipoprotein in plasma and is secreted in mucosal surfaces, allowing for its detection in nasal fluid lavage. In this study, ApoA1 has demonstrated antimicrobial activity and an essential role as a mucosal immune modulator, regulating cytokine production. It exhibits anti-inflammatory and antioxidant activities.	[62]
Annexin A1	Functional-regulatory protein. This protein is a calcium-dependent phospholipid-binding protein known to have anti-inflammatory activity and is involved in modulating the host inflammatory response in chronic polypoid nasal inflammation.	[62]
Catalase	Functional-regulatory protein. This protein is an antioxidant enzyme that protects the cells against damage from oxidative stress by converting hydrogen peroxide into oxygen and water. Diminished abundance of this protectant enzyme in the sinonasal mucosa maybe a contributing factor in the marked cellular damage that characterizes CRSwNP.	[62]
Protein S100 subtype A8	Functional-regulatory protein. This protein is a cellular regulatory protein involved in host defense in the respiratory epithelium, in which it exhibits a proinflammatory role as a neutrophil attractant.	[62]
Phosphatidylethanolamine-binding protein 1	Functional-regulatory protein. It is a phospholipid-binding protein localized to the cytoplasm and plasma membrane in respiratory epithelial cells and acts as an endogenous activator of the mitogen-activated protein kinase (MAPK) apoptotic pathway in epithelial cells.	[62]
RHO-GDP dissociation inhibitor 2 (RHO-GD2)	Functional-regulatory protein. The antiapoptotic activity of this cellular regulatory protein, in association with its increased abundance in polypoid sinonasal tissue in multiple studies, suggests that it may play a significant role in polyp formation.	[62]
Keratin type II-8	Cytoskeletal protein. This protein is an intermediate filament protein that protects the epithelial cells from mechanical and nonmechanical stresses and thus from cell death.	[62]
Actin, cytoplasmic-1, cytoplasmic-2	Cytoskeletal protein. The significant difference in the abundance of 2 cytoskeletal actin isoforms, A1 and A2, highlights the complex nature of reorganization occurring at the cellular level in chronic inflammatory disease (CRSwNP).	[62]
Lysozyme C precursor	A protein of macrophage origin. This protein, which was expressed less in CRS patients, exhibits a bacteriolytic function. It is generally associated with the monocyte–macrophage system and enhances the activity of immunogens.	[56,63,64]
Clara cell phospholipid-binding protein (CCPBP)	A protein of glandular cell origin. This protein is a potent inhibitor of phospholipase A2, an enzyme that is crucial in initiating the metabolism of arachidonic acid into leukotrienes and prostaglandins. It was also significantly underexpressed in disease patients.	[63,64]
Antileukoproteinase 1 precursor (ALP; HUSI-1; seminal proteinase inh)	A protein of glandular cell origin. This protein is an acid-stable proteinase inhibitor secreted in mucous fluids. The pathophysiology of several respiratory tract diseases is related to an imbalance between inflammatory cell proteases and their inhibitors.	[63,64]
Hypothetical protein IGLV3-21	A protein of unknown origin. The function of the hypothetical protein Q6NS96 is unknown. Its lesser expression in CRS patients suggests that it may play a protective role against this disease.	[63]
IGLV4-3 (immunoglobulin lambda variable 4-3)	A protein of unknown origin. The origin and function of this protein are unknown. It may play a role in the pathogenesis of CRS or act as a marker of inflammation within the nasal cavity and paranasal sinuses.	[64]
Bactericidal protein	A protein of glandular cell origin. This protein has been demonstrated to be important in the host defense mechanism against bacterial organisms and was underexpressed in CRS patients.	[63,64]
Calgranulin	A protein of macrophage origin. This protein is overexpressed in CRS patients and known to be expressed by macrophages in acutely and chronically inflamed tissues. Bactericidal/permeability-increasing protein essential in the host defense against bacteria, and it may play a role in the pathogenesis of CRS or act as a marker of inflammation within the nasal cavity and paranasal sinuses.	[63,64]
Lipocalin	A protein of epithelial cell origin. This protein was moderately overexpressed in CRS patients, and its significance is not clear. Research on tear lipocalin has shown that it is induced by infection or inflammation, suggesting that it may be involved in the physiologic protection of the factor of epithelial tissue in vivo.	[56,63,64]
Amylase α1A (AMY1A)	A protein of glandular cell origin. This protein was found to be markedly overexpressed in CRS patients. It may probably play a role in the pathogenesis of CRS or act as a marker of inflammation within the nasal cavity and paranasal sinuses.	[63,64]

Abbreviations: CF = cystic fibrosis; CRS = chronic rhinosinusitis; CRSwNP = chronic rhinosinusitis with nasal polyps; NLF = nasal lavage fluid.

**Table 3 jpm-10-00245-t003:** Metabolites found to be potentially relevant in CRS development.

Metabolites	Strategy	Biological Sample	Number of Cases/Controls	Analytical Technique	Sample Pretreatment	References
- Palmitic acid, oleic acid, stearic acid, lauric acid - Phosphatidylcholines - Phosphatidylethanolamines - Ceramides and cholesteryl esters - Cholesteryl palmitoleate present only in CRSwNP - Cholesteryl arachidonate, cholesteryl linoleate	Untargeted lipidomics	Sinus mucosa	9 CRSwNP, 11 CRSsNP, 12 controls	HPTLC, GC-EI/MS, and FI/ESI-MS/MS	Bligh-Dyer lipid extraction	[75]
- Prostaglandin E2, prostaglandin D2, thromboxane B2, 12-hydroxyheptadecatrienoic acid - 11-Hydroxyeicosatetraenoic acid - 12-Hydroxyeicosatetraenoic acid, 15-hydroxyeicosatetraenoic acid, lipoxin A4, 8, 15-dihete, and 17-hydroxydocosahexaenoic acid	Targeted lipidomics	Cells isolated from nasal polyps	6 patients with eosinophilic rhinosinusitis, 6 controls	LC-MS/MS	Anion-exchange SPE	[77]
- Neopterin	Targeted study	Serum and nasal secretions	54 CRSwNP, 46 CRSsNP, 40 controls, 51 AR, 43 NAR	Immunoassay	Dilution and centrifugation	[78]
- VOCs without exact identification	Untargeted metabolomics	Exhaled breath samples	17 CRS 24 NP, 30 controls	Technique based on the use of cross-reactive nanoarray with various sensors	Thermal desorption of breath trapped in nanotubes	[81]
- Exhaled nasal nitric oxide	Targeted study	Exhaled breath samples	54 CRSwNP 34 CRSsNP 20 controls	NIOX device	N/A	[82]

Abbreviations: AR = allergic rhinitis; CRS = chronic rhinosinusitis; CRSsNP = chronic rhinosinusitis without nasal polyps; CRSwNP = chronic rhinosinusitis with nasal polyps; FI/ESI-MS/MS = flow-injection/electrospray ionization-tandem mass spectrometry; GC-EI/MS = gas chromatography/electron impact-mass spectrometry; HPTLC = high-performance thin-layer chromatography; LC-MS/MS = liquid chromatography–tandem mass spectrometry; NAR = nonallergic rhinopathy; N/A = not applicable; NP = nasal polyps; SPE = solid-phase extraction; VOCs = volatile organic compounds.

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
