# Peer review of "An Overview of the Application of Systems Biology in an Understanding of Chronic Rhinosinusitis (CRS) Development"

_jpm, 2020, doi:10.3390/jpm10040245_

Round 1
Reviewer 1 Report
Unfortunately I cannot seem to access a version where the changes are tracked making re-review a little challenging.
I can see however that the manuscript has been extensively reworked and from what I can see the spelling and grammar has been improved.
Reviewer 2 Report
The proposed review manuscript by Michalik et al. summarizes the current knowledge on the application of systems biology in the understanding of chronic rhinosinusitis (CRS) development based on genomic, transcriptomic, proteomic, and metabolomic studies.
In my opinion, the review is very well structured and it is informative. I do not have any suggestions for improvement.
This manuscript is a resubmission of an earlier submission. The following is a list of the peer review reports and author responses from that submission.
Round 1
Reviewer 1 Report
The present review tries to provide an overview of the application of systems biology approaches to the study of CRS. However, the manuscript presents several flaws that should be addressed:
- The writing of the review should be improved in order to catch reader’s attention. I would suggest to improve the writing, in a more scientific way. Moreover, there are some English grammatical errors that should be corrected (see for example, lines 89-90, line 119: ‘genes are then individually genotyped’, line 207: ‘with ad without’, line 237: ‘much as’, line 239: ‘certain point in time’, and many more).
- Although the review may be interesting, the narrative should be improved and provide the reader a holistic view of the studies that have used omics data in the study of CRS. The review is a list of results of the studies performed in this area, but an integration and a summary of the different results is lacking. Therefore, I would recommend the authors to provide a summary table of the results obtained in the application of each omics data, similarly to what they did in Table 1. For example, a Table summarizing the genetic variants found to be associated with CRS, their genomic location, gene annotation, risk allele, study design, number of cases and controls (if it is the case), and the reference of the study would be very helpful. Similar tables including the main findings of studies using proteomics or metabolomics would be recommended. Those tables should also include the biological samples used in each study.
- There are some concepts that are used in an inappropriate way. For example, authors used genetic code when they mean genetic information. Similarly, the term ‘Systemic biology’ should be replaced with Systems biology.
- I would suggest to remove the Figures 3, 4, 5, and 6, and substitute them with the summary tables I mentioned before. Those figures can be found in their original papers.
- Authors should also include studies that integrate several omics data if any, as well as discuss how to integrate several type of data.
- I missed studies on the relationship between the microbiome of the nasal mucus in CRS.
- The conclusions of the review should give an overview of the studies conducted so far and their main findings in a more holistic and integrative way. They may include some recommendations and how the authors envisage future studies in this topic.
Author Response
Response to Reviewer 1 Comments
1. The writing of the review should be improved in order to catch reader’s attention. I would suggest to improve the writing, in a more scientific way. Moreover, there are some English grammatical errors that should be corrected (see for example, lines 89-90, line 119: ‘genes are then individually genotyped’, line 207: ‘with ad without’, line 237: ‘much as’, line 239: ‘certain point in time’, and many more).
Response 1: Thank you for your suggestion. We did an extensive revision of our manuscript and correct all the mistakes and shortcomings. According to the guidelines, any revisions were highlighted, using the "Track Changes" function in Microsoft Word.
2. Although the review may be interesting, the narrative should be improved and provide the reader a holistic view of the studies that have used omics data in the study of CRS. The review is a list of results of the studies performed in this area, but an integration and a summary of the different results is lacking. Therefore, I would recommend the authors to provide a summary table of the results obtained in the application of each omics data, similarly to what they did in Table 1. For example, a Table summarizing the genetic variants found to be associated with CRS, their genomic location, gene annotation, risk allele, study design, number of cases and controls (if it is the case), and the reference of the study would be very helpful. Similar tables including the main findings of studies using proteomics or metabolomics would be recommended. Those tables should also include the biological samples used in each study.
Response 2: Thank you for your suggestion. According to your suggestions, we add two tables that summarize the data reported in genomics and metabolomics studies. They are now in Table 1 (genomics) and table 3 (metabolomics) in the revised version of the manuscript. Data from proteomics already were in the first version of the manuscript and still are in a revised version of the manuscript as a table 2.
3. There are some concepts that are used in an inappropriate way. For example, authors used genetic code when they mean genetic information. Similarly, the term ‘Systemic biology’ should be replaced with Systems biology.
Response 3: Thank you very much for your remark. We did all the necessary corrections according to your suggestion and there are no more 'genetic code' and 'systemic biology' present in the manuscript.
4. I would suggest to remove the Figures 3, 4, 5, and 6, and substitute them with the summary tables I mentioned before. Those figures can be found in their original papers.
Response 4: We agree with the suggestions of the reviewer and decided to remove two figures numbered 3 and 5. Figures 4 and 6 we decided to leave in the revised version of the manuscript since we believe that they are presenting a close look at the details of the cited study. The two summary tables suggested by the reviewer have been placed instead of these two figures, however if the editor and reviewer insist on the removal of the other two figures from the manuscript we would do it without hesitation.
5. Authors should also include studies that integrate several omics data if any, as well as discuss how to integrate several type of data.
Response 5: Thank you for this suggestion. Unfortunately, based on our best knowledge, no single study integrates all or only two of the -omics related fields. Therefore we did not improve a text of the manuscript on that aspect of research however, we strongly believe, and that was also our motivation to write this manuscript, that future research in CRS should concentrate on integrating knowledge from different levels of the cell's organization, resulting in a better understanding of the disease pathogenesis and new directions of pharmacotherapy.
6. I missed studies on the relationship between the microbiome of the nasal mucus in CRS.
Response 6: We have decided not to write about studies on the microbiota and the nasal mucus in CRS since that is a very broad field of research which, in our opinion, expand the size of this review. We fully agree with a reviewer that this is an important field that is worthy of study and should be comprehensively described but to keep the text of this review consistent we have omitted this field of research. However, when the editor and reviewer find it necessary to add research on the microbiome and its relationships with the nasal mucus in CRS to this review we will be able to complete its expanding size of the manuscript.
7. The conclusions of the review should give an overview of the studies conducted so far and their main findings in a more holistic and integrative way. They may include some recommendations and how the authors envisage future studies in this topic.
Response 7: Thank you for this suggestion. After careful reading of the manuscript, we found that concussions truly needs rewriting. We do hope that in its current form it would be more suitable for this review manuscript.
Reviewer 2 Report
Thanks for the opportunity to read your manuscript.
I found this to be relevant, comprehensive and interesting.
Figure 2 is hard to read - do you have a copy with better resolution?
My major criticism however relates to the text: not only the English grammar but frequency typos and variable use of abbreviations e.g. NLF is defined in line 250. Later there are references to NFL (line 317), NLF is then defined again in line 341 and then there is reference to NLP (presumably meaning NLF) in lines 366 and 368.
In total on my single read through I found 14 typos or errors of grammar and I suspect there are more:
- It has been demonstrated in chronic rhinosinusitis presents of cytokine protein
- become identifiable, identifiable and also for interpretation
- Numerous factors can influence chronic rhinosinusitis development, among which genetic predisposition should be underlined
- there is also evidences that primary ciliary dyskinesia
- pGWASstudy
- compared DNA pools from patients with ad without
- canonical pathways (B) were generatedusing
- transcriptomicfactors
- involved in acquired and innate immune as a result of factors
- Some interesting of them are shown in the table 1.
- a lotof the significantly elevated proteins
- The metabolic fingerprinting is kind of qualitative study
- incubated at centrifuged
- docosanoidswere
Author Response
Response to Reviewer 2 Comments
Point 1:
My major criticism however relates to the text: not only the English grammar but frequency typos and variable use of abbreviations e.g. NLF is defined in line 250. Later there are references to NFL (line 317), NLF is then defined again in line 341 and then there is reference to NLP (presumably meaning NLF) in lines 366 and 368.
In total on my single read through I found 14 typos or errors of grammar and I suspect there are more:
- It has been demonstrated in chronic rhinosinusitis presents of cytokine protein
- become identifiable, identifiable and also for interpretation
- Numerous factors can influence chronic rhinosinusitis development, among which genetic predisposition should be underlined
- there is also evidences that primary ciliary dyskinesia
- pGWASstudy
- compared DNA pools from patients with ad without
- canonical pathways (B) were generatedusing
- transcriptomicfactors
- involved in acquired and innate immune as a result of factors
- Some interesting of them are shown in the table 1.
- a lotof the significantly elevated proteins
- The metabolic fingerprinting is kind of qualitative study
- incubated at centrifuged
- docosanoidswere
Response 1: Thank you for your comment and indication of our mistakes. We have revised our manuscript and correct all the mistakes and shortcomings according to your suggestions and English grammar. According to the guidelines, any revisions were highlighted, using the "Track Changes" function in Microsoft Word.